# LARGE-WIDTH FUNCTIONAL ASYMPTOTICS FOR DEEP GAUSSIAN NEURAL NETWORKS

**Daniele Bracale**[1], **Stefano Favaro**[1,2], **Sandra Fortini**[3], **Stefano Peluchetti**[4]
[1] University of Torino, [2] Collegio Carlo Alberto, [3] Bocconi University, [4] Cogent Labs

## ABSTRACT

In this paper, we consider fully-connected feed-forward deep neural networks where weights and biases are independent and identically distributed according to Gaussian distributions. Extending previous results (Matthews et al., 2018a;b; Yang, 2019) we adopt a function-space perspective, i.e. we look at neural networks as infinite-dimensional random elements on the input space $\mathbb{R}^I$. Under suitable assumptions on the activation function we show that: i) a network defines a continuous stochastic process on the input space $\mathbb{R}^I$; ii) a network with re-scaled weights converges weakly to a continuous Gaussian Process in the large-width limit; iii) the limiting Gaussian Process has almost surely locally $\gamma$-Hölder continuous paths, for $0 < \gamma < 1$. Our results contribute to recent theoretical studies on the interplay between infinitely-wide deep neural networks and Gaussian Processes by establishing weak convergence in function-space with respect to a stronger metric.

## 1 INTRODUCTION

The interplay between infinitely-wide deep neural networks and classes of Gaussian Processes has its origins in the seminal work of Neal (1995), and it has been the subject of several theoretical studies. See, e.g., Der & Lee (2006), Lee et al. (2018), Matthews et al. (2018a;b), Yang (2019) and references therein. Let consider a fully-connected feed-forward neural network with re-scaled weights composed of $L \geq 1$ layers of widths $n_1, \ldots, n_L$, i.e.

$$
\begin{aligned}
f_i^{(1)}(x) &= \sum_{j=1}^{I} w_{i,j}^{(1)} x_j + b_i^{(1)} & i &= 1, \ldots, n_1 \\
f_i^{(l)}(x) &= \frac{1}{\sqrt{n_{l-1}}} \sum_{j=1}^{n_{l-1}} w_{i,j}^{(l)} \phi(f_j^{(l-1)}(x)) + b_i^{(l)} & l &= 2, \ldots, L, \ \ i = 1, \ldots, n_l
\end{aligned}
\tag{1}
$$

where $\phi$ is a non-linearity and $x \in \mathbb{R}^I$ is a real-valued input of dimension $I \in \mathbb{N}$. Neal (1995) considered the case $L = 2$, a finite number $k \in \mathbb{N}$ of fixed distinct inputs $(x^{(1)}, \ldots, x^{(k)})$, with each $x^{(r)} \in \mathbb{R}^I$, and weights $w_{i,j}^{(l)}$ and biases $b_i^{(l)}$ independently and identically distributed (iid) as Gaussian distributions. Under appropriate assumptions on the activation $\phi$ Neal (1995) showed that: i) for a fixed unit $i$, the $k$-dimensional random vector $(f_i^{(2)}(x^{(1)}), \ldots, f_i^{(2)}(x^{(k)}))$ converges in distribution, as the width $n_1$ goes to infinity, to a $k$-dimensional Gaussian random vector; ii) the large-width convergence holds jointly over finite collections of $i$'s and the limiting $k$-dimensional Gaussian random vectors are independent across the index $i$. These results concerns neural networks with a single hidden layer, but Neal (1995) also includes preliminary considerations on infinitely-wide deep neural networks. More recent works, such as Lee et al. (2018), established convergence results corresponding to Neal (1995) results i) and ii) for deep neural networks under the assumption that widths $n_1, \ldots, n_L$ go to infinity sequentially over network layers. Matthews et al. (2018a;b) extended the work of Neal (1995); Lee et al. (2018) by assuming that the width $n$ grows to infinity jointly over network layers, instead of sequentially, and by establishing joint convergence over all $i$ and countable distinct inputs. The joint growth over the layers is certainly more realistic than the sequential growth, since the infinite Gaussian limit is considered as an approximation of a very wide network. We operate in the same setting of Matthews et al. (2018b), hence from here onward $n \geq 1$ denotes the

common layer width, i.e. $n_1, \ldots, n_L = n$. Finally, similar large-width limits have been established for a great variety of neural network architectures, see for instance Yang (2019).

The assumption of a countable number of fixed distinct inputs is the common trait of the literature on large-width asymptotics for deep neural networks. Under this assumption, the large-width limit of a network boils down to the study of the large-width asymptotic behavior of the $k$-dimensional random vector $(f_i^{(l)}(x^{(1)}), \ldots, f_i^{(l)}(x^{(k)}))$ over $i \geq 1$ for finite $k$. Such limiting finite-dimensional distributions describe the large-width distribution of a neural network a priori over any dataset, which is finite by definition. When the limiting distribution is Gaussian, as it often is, this immediately paves the way to Bayesian inference for the limiting network. Such an approach is competitive with the more standard stochastic gradient descent training for the fully-connected architectures object of our study (Lee et al., 2020). However, knowledge of the limiting finite-dimensional distributions is not enough to infer properties of the limiting neural network which are inherently uncountable such as the continuity of the limiting neural network, or the distribution of its maximum over a bounded interval. Results in this direction give a more complete understanding of the assumptions being made a priori, and hence whether a given model is appropriate for a specific application. For instance, Van Der Vaart & Van Zanten (2011) shows that for Gaussian Processes the function smoothness under the prior should match the smoothness of the target function for satisfactory inference performance.

In this paper we thus consider a novel, and more natural, perspective to the study of large-width limits of deep neural networks. This is an infinite-dimensional perspective where, instead of fixing a countable number of distinct inputs, we look at $f_i^{(l)}(x, n)$ as a stochastic process over the input space $\mathbb{R}^I$. Under this perspective, establishing large-width limits requires considerable care and, in addition, it requires to show the existence of both the stochastic process induced by the neural network and its large-width limit. We start by proving the existence of i) a continuous stochastic process, indexed by the network width $n$, corresponding to the fully-connected feed-forward deep neural network; ii) a continuous Gaussian Process corresponding to the infinitely-wide limit of the deep neural network. Then, we prove that the stochastic process i) converges weakly, as the width $n$ goes to infinity, to the Gaussian Process ii) jointly over all units $i$. As a by-product of our results, we show that the limiting Gaussian Process has almost surely locally $\gamma$-Hölder continuous paths, for $0 < \gamma < 1$. To make the exposition self-contained we include an alternative proof of the main result of Matthews et al. (2018a;b), i.e. the finite-dimensional limit for full-connected neural networks. The major difference between our proof and that of Matthews et al. (2018b) is due to the use of the characteristic function to establish convergence in distribution, instead of relying on a CLT (Blum et al., 1958) for exchangeable sequences.

The paper is structured as follows. In Section 2 we introduce the setting under which we operate, whereas in Section 3 we present a high-level overview of the approach taken to establish our results. Section 4 contains the core arguments of the proof of our large-width functional limit for deep Gaussian neural networks, which are spelled out in detail in the supplementary material (SM). We conclude in Section 5.

## 2 SETTING

Let $(\Omega, \mathcal{H}, \mathbb{P})$ be the probability space on which all random elements of interest are defined. Furthermore, let $N(\mu, \sigma^2)$ denote a Gaussian distribution with mean $\mu \in \mathbb{R}$ and strictly positive variance $\sigma^2 \in \mathbb{R}_+$, and let $N_k(\mathbf{m}, \Sigma)$ be a $k$-dimensional Gaussian distribution with mean $\mathbf{m} \in \mathbb{R}^k$ and covariance matrix $\Sigma \in \mathbb{R}^{k \times k}$. In particular, $\mathbb{R}^k$ is equipped with $\|\cdot\|_{\mathbb{R}^k}$, the euclidean norm induced by the inner product $\langle \cdot, \cdot \rangle_{\mathbb{R}^k}$, and $\mathbb{R}^\infty = \times_{i=1}^\infty \mathbb{R}$ is equipped with $\|\cdot\|_{\mathbb{R}^\infty}$, the norm induced by the distance $d(\mathbf{a}, \mathbf{b})_\infty = \sum_{i \geq 1} \xi(|a_i - b_i|)/2^i$ for $\mathbf{a}, \mathbf{b} \in \mathbb{R}^\infty$ (Theorem 3.38 of Aliprantis & Border (2006)), where $\xi(t) = t/(1+t)$ for all real values $t \geq 0$. Note that $(\mathbb{R}, |\cdot|)$ and $(\mathbb{R}^\infty, \|\cdot\|_{\mathbb{R}^\infty})$ are Polish spaces, i.e. separable and complete metric spaces (Corollary 3.39 of Aliprantis & Border (2006)). We choose $d_\infty$ since it generates a topology that coincides with the product topology (line 5 of the proof of Theorem 3.36 of Aliprantis & Border (2006)). The space $(S, d)$ will indicate a generic Polish space such as $\mathbb{R}$ or $\mathbb{R}^\infty$ with the associated distance. We indicate with $S^{\mathbb{R}^I}$ the space of functions from $\mathbb{R}^I$ into $S$ and $C(\mathbb{R}^I; S) \subset S^{\mathbb{R}^I}$ the space of continuous functions from $\mathbb{R}^I$ into $S$.

Let $\omega_{i,j}^{(l)}$ be the random weights of the $l$-th layer, and assume that they are iid as $N(0, \sigma_\omega^2)$, i.e.

$$\varphi_{\omega_{i,j}^{(l)}}(t) = \mathbb{E}[e^{it\omega_{i,j}^{(l)}}] = e^{-\frac{1}{2}\sigma_\omega^2 t^2} \tag{2}$$

is the characteristic function of $\omega_{i,j}^{(l)}$, for $i \geq 1$, $j = 1, \ldots, n$ and $l \geq 1$. Let $b_i^{(l)}$ be the random biases of the $l$-th layer, and assume that they are iid as $N(0, \sigma_b^2)$, i.e.

$$\varphi_{b_i^{(l)}}(t) = \mathbb{E}[e^{itb_i^{(l)}}] = e^{-\frac{1}{2}\sigma_b^2 t^2} \tag{3}$$

is the characteristic function of $b_i^{(l)}$, for $i \geq 1$ and $l \geq 1$. Weights $\omega_{i,j}^{(l)}$ are independent of biases $b_i^{(l)}$, for any $i \geq 1$, $j = 1, \ldots, n$ and $l \geq 1$. Let $\phi : \mathbb{R} \to \mathbb{R}$ denote a continuous non-linearity. For the finite-dimensional limit we will assume the polynomial envelop condition

$$|\phi(s)| \leq a + b|s|^m, \tag{4}$$

for any $s \in \mathbb{R}$ and some real values $a, b > 0$ and $m \geq 1$. For the functional limit we will use a stronger assumption on $\phi$, assuming $\phi$ to be Lipschitz on $\mathbb{R}$ with Lipschitz constant $L_\phi$.

Let $Z$ be a stochastic process on $\mathbb{R}^I$, i.e. for each $x \in \mathbb{R}^I$, $Z(x)$ is defined on $(\Omega, \mathcal{H}, \mathbb{P})$ and it takes values in $S$. For any $k \in \mathbb{N}$ and $x_1, \ldots, x_k \in \mathbb{R}^I$, let $P_{x_1, \ldots, x_k}^Z = \mathbb{P}(Z(x_1) \in A_1, \ldots, Z(x_k) \in A_k)$, with $A_1, \ldots, A_k \in \mathcal{B}(S)$. Then, the family of finite-dimensional distributions of $Z(x)$ is defined as the family of distributions $\{P_{x_1, \ldots, x_k}^Z : x_1, \ldots, x_k \in \mathbb{R}^I \text{ and } k \in \mathbb{N}\}$. See, e.g., Billingsley (1995). In Definition 1 and Definition 2 we look at the deep neural network (1) as a stochastic process on input space $\mathbb{R}^I$, that is a stochastic process whose finite-dimensional distributions are determined by a finite number $k \in \mathbb{N}$ of fixed distinct inputs $(x^{(1)}, \ldots, x^{(k)})$, with each $x^{(r)} \in \mathbb{R}^I$. The existence of the stochastic processes of Definition 1 and Definition 2 will be thoroughly discussed in Section 3.

**Definition 1.** *For any fixed $l \geq 2$ and $i \geq 1$, let $(f_i^{(l)}(n))_{n \geq 1}$ be a sequence of stochastic processes on $\mathbb{R}^I$. That is, $f_i^{(l)}(n) : \mathbb{R}^I \to \mathbb{R}$, with $x \mapsto f_i^{(l)}(x, n)$, is a stochastic process on $\mathbb{R}^I$ whose finite-dimensional distributions are the laws, for any $k \in \mathbb{N}$ and $x^{(1)}, \ldots, x^{(k)} \in \mathbb{R}^I$, of the $k$-dimensional random vectors*

$$f_i^{(1)}(\boldsymbol{X}, n) = f_i^{(1)}(\boldsymbol{X}) = [f_i^{(1)}(x^{(1)}, n), \ldots, f_i^{(1)}(x^{(k)}, n)]^T = \sum_{j=1}^{I} \omega_{i,j}^{(1)} \boldsymbol{x}_j + b_i^{(1)} \boldsymbol{1} \tag{5}$$

$$f_i^{(l)}(\boldsymbol{X}, n) = [f_i^{(l)}(x^{(1)}, n), \ldots, f_i^{(l)}(x^{(k)}, n)]^T = \frac{1}{\sqrt{n}} \sum_{j=1}^{n} \omega_{i,j}^{(l)}(\phi \bullet f_j^{(l-1)}(\boldsymbol{X}, n)) + b_i^{(l)} \boldsymbol{1} \tag{6}$$

*where $\boldsymbol{X} = [x^{(1)}, \ldots, x^{(k)}] \in \mathbb{R}^{I \times k}$ is a $I \times k$ input matrix of $k$ distinct inputs $x^{(r)} \in \mathbb{R}^I$, $\boldsymbol{1}$ denotes a vector of dimension $k \times 1$ of $1$'s, $\boldsymbol{x}_j$ denotes the $j$-th row of the input matrix and $\phi \bullet \boldsymbol{X}$ is the element-wise application of $\phi$ to the matrix $\boldsymbol{X}$. Let $f_{r,i}^{(l)}(\boldsymbol{X}, n) = \boldsymbol{1}_r^T f_i^{(l)}(\boldsymbol{X}, n) = f_i^{(l)}(x^{(r)}, n)$ denote the $r$-th component of the $k \times 1$ vector $f_i^{(l)}(\boldsymbol{X}, n)$, being $\boldsymbol{1}_r$ a vector of dimension $k \times 1$ with $1$ in the $r$-the entry and $0$ elsewhere.*

**Remark**: in contrast to (1), we have defined (5)-(6) over an infinite number of units $i \geq 1$ over each layer $l$, but the dependency on each previous layer $l-1$ remains limited to the first $n$ components.

**Definition 2.** *For any fixed $l \geq 2$, let $(\boldsymbol{F}^{(l)}(n))_{n \geq 1}$ be a sequence of stochastic processes on $\mathbb{R}^I$. That is, $\boldsymbol{F}^{(l)}(n) : \mathbb{R}^I \to \mathbb{R}^\infty$, with $x \mapsto \boldsymbol{F}^{(l)}(x, n)$, is a stochastic process on $\mathbb{R}^I$ whose finite-dimensional distributions are the laws, for any $k \in \mathbb{N}$ and $x^{(1)}, \ldots, x^{(k)} \in \mathbb{R}^I$, of the $k$-dimensional random vectors*

$$\begin{cases} \boldsymbol{F}^{(1)}(\boldsymbol{X}) = \left[ f_1^{(1)}(\boldsymbol{X}), f_2^{(1)}(\boldsymbol{X}), \ldots \right]^T \\ \boldsymbol{F}^{(l)}(\boldsymbol{X}, n) = \left[ f_1^{(l)}(\boldsymbol{X}, n), f_2^{(l)}(\boldsymbol{X}, n), \ldots \right]^T. \end{cases}$$

**Remark**: for $k$ inputs, the vector $\boldsymbol{F}^{(l)}(\boldsymbol{X}, n)$ is an $\infty \times k$ array, and for a single input $x^{(r)}$, $\boldsymbol{F}^{(l)}(x^{(r)}, n)$ can be written as $[f_1^{(l)}(x^{(r)}, n), f_2^{(l)}(x^{(r)}, n), \ldots]^T \in \mathbb{R}^{\infty \times 1}$. We define $\boldsymbol{F}_r^{(l)}(\boldsymbol{X}, n) = \boldsymbol{F}^{(l)}(x^{(r)}, n)$ the $r$-th column of $\boldsymbol{F}^{(l)}(\boldsymbol{X}, n)$. When we write $\langle \boldsymbol{F}^{(l-1)}(x, n), \boldsymbol{F}^{(l-1)}(y, n) \rangle_{\mathbb{R}^n}$ (see (8)) we treat $\boldsymbol{F}^{(l)}(x, n)$ and $\boldsymbol{F}^{(l)}(y, n)$ as elements in $\mathbb{R}^n$ and not in $\mathbb{R}^\infty$, i.e. we consider only the first $n$ components of $\boldsymbol{F}^{(l)}(x, n)$ and $\boldsymbol{F}^{(l)}(y, n)$.

## 3  PLAN SKETCH

We start by recalling the notion of convergence in law, also referred to as convergence in distribution or weak convergence, for a sequence of stochastic processes. See Billingsley (1995) for a comprehensive account.

**Definition 3** (convergence in distribution). *Suppose that $f$ and $(f(n))_{n\geq 1}$ are random elements in a topological space $C$. Then, $(f(n))_{n\geq 1}$ is said to converge in distribution to $f$, if $\mathbb{E}[h(f(n))] \to \mathbb{E}[h(f)]$ as $n \to \infty$ for every bounded and continuous function $h : C \to \mathbb{R}$. In that case we write $f(n) \xrightarrow{d} f$.*

In this paper, we deal with continuous and real-valued stochastic processes. More precisely, we consider random elements defined on $C(\mathbb{R}^I; S)$, with $(S, d)$ Polish space. Our aim is to study in $C(\mathbb{R}^I; S)$ the convergence in distribution as the width $n$ goes to infinity for:

i) the sequence $(f_i^{(l)}(n))_{n\geq 1}$ for a fixed $l \geq 2$ and $i \geq 1$ with $(S, d) = (\mathbb{R}, |\cdot|)$, i.e. the neural network process for a single unit;

ii) the sequence $(\mathbf{F}^{(l)}(n))_{n\geq 1}$ for a fixed $l \geq 2$ with $(S, d) = (\mathbb{R}^\infty, \|\cdot\|_\infty)$, i.e. the neural network process for all units.

Since applying Definition 3 in a function space is not easy, we need, proved in SM F, the following proposition.

**Proposition 1** (convergence in distribution in $C(\mathbb{R}^I; S)$, $(S, d)$ Polish). *Suppose that $f$ and $(f(n))_{n\geq 1}$ are random elements in $C(\mathbb{R}^I; S)$ with $(S, d)$ Polish space. Then, $f(n) \xrightarrow{d} f$ if: i) $f(n) \xrightarrow{f_d} f$ and ii) the sequence $(f(n))_{n\geq 1}$ is uniformly tight.*

We denoted with $\xrightarrow{f_d}$ the convergence in law of the finite-dimensional distributions of a sequence of stochastic processes. The notion of tightness formalizes the concept that the probability mass is not allowed to "escape at infinity": a single random element $f$ in a topological space $C$ is said to be tight if for each $\epsilon > 0$ there exists a compact $T \subset C$ such that $\mathbb{P}[f \in C \setminus T] < \epsilon$. If a metric space $(C, \rho)$ is Polish any random element on the Borel $\sigma$-algebra of $C$ is tight. A sequence of random elements $(f(n))_{n\geq 1}$ in a topological space $C$ is said to be uniformly tight[1] if for every $\epsilon > 0$ there exists a compact $T \subset C$ such that $\mathbb{P}[f(n) \in C \setminus T] < \epsilon$ for all $n$.

According to Proposition 1, to achieve convergence in distribution in function spaces we need the following Steps A-D:

**Step A)** to establish the existence of the finite-dimensional weak-limit $f$ on $\mathbb{R}^I$. We will rely on Theorem 5.3 of Kallenberg (2002), known as Levy theorem.

**Step B)** to establish the existence of the stochastic processes $f$ and $(f(n))_{n\geq 1}$ as elements in $S^{\mathbb{R}^I}$ the space of function from $\mathbb{R}^I$ into $S$. We make use of Daniell-Kolmogorov criterion (Kallenberg, 2002, Theorem 6.16): given a family of multivariate distributions $\{P_\mathcal{I}$ probability measure on $\mathbb{R}^{\dim(\mathcal{I})} \mid \mathcal{I} \subset \{x^{(1)}, \dots, x^{(k)}\}_{x^{(z)} \in \mathbb{R}^I, k \in \mathbb{N}}\}$ there exists a stochastic process with $\{P_\mathcal{I}\}$ as finite-dimensional distributions if $\{P_\mathcal{I}\}$ satisfies the projective property: $P_J(\cdot \times \mathbb{R}_{J \setminus \mathcal{I}}) = P_\mathcal{I}(\cdot), \mathcal{I} \subset J \subset \{x^{(1)}, \dots, x^{(k)}\}_{x^{(z)} \in \mathbb{R}^I, k \in \mathbb{N}}$. That is, it is required consistency with respect to the marginalization over arbitrary components. In this step we also suppose, for a moment, that the stochastic processes $(f(n))_{n\geq 1}$ and $f$ belong to $C(\mathbb{R}^I; S)$ and we establish the existence of such stochastic processes in $C(\mathbb{R}^I; S)$ endowed with a $\sigma$-algebra and a probability measure that will be defined.

**Step C)** to show that the stochastic processes $(f(n))_{n\geq 1}$ and $f$ belong to $C(\mathbb{R}^I; S) \subset S^{\mathbb{R}^I}$. With regards to $(f(n))_{n\geq 1}$ this is a direct consequence of (5)-(6) and the continuity of $\phi$. With regards to the limiting process $f$, with an additional Lipschitz assumption on $\phi$, we rely on the following Kolmogorov-Chentsov criterion (Kallenberg, 2002, Theorem 3.23):

---

[1] Kallenberg (2002) uses the same term "tightness" for both cases of a single random element and of sequences of random elements; we find that the introduction of "uniform tightness" brings more clarity.

**Proposition 2** (continuous version and local-Hölderianity, $(S, d)$ complete). *Let $f$ be a process on $\mathbb{R}^I$ with values in a complete metric space $(S, d)$, and assume that there exist $a, b, H > 0$ such that,*

$$\mathbb{E}[d(f(x), f(y))^a] \leq H\|x - y\|^{(I+b)}, \quad x, y \in \mathbb{R}^I$$

*Then $f$ has a continuous version (i.e. $f$ belongs to $C(\mathbb{R}^I; S)$), and the latter is a.s. locally Hölder continuous with exponent $c$ for any $c \in (0, b/a)$.*

**Step D)** the uniform tightness of $(f(n))_{n \geq 1}$ in $C(\mathbb{R}^I; S)$. We rely on an extension of the Kolmogorov-Chentsov criterion (Kallenberg, 2002, Corollary 16.9), which is stated in the following proposition.

**Proposition 3** (uniform tightness in $C(\mathbb{R}^I; S)$, $(S, d)$ Polish). *Suppose that $(f(n))_{n \geq 1}$ are random elements in $C(\mathbb{R}^I; S)$ with $(S, d)$ Polish space. Assume that $f(0_{\mathbb{R}^I}, n)_{n \geq 1}$ (i.e. $f(n)$ evaluated at the origin) is uniformly tight in $S$ and that there exist $a, b, H > 0$ such that,*

$$\mathbb{E}[d(f(x, n), f(y, n))^a] \leq H\|x - y\|^{(I+b)}, \quad x, y \in \mathbb{R}^I, n \in \mathbb{N}$$

*uniformly in $n$. Then $(f(n))_{n \geq 1}$ is uniformly tight in $C(\mathbb{R}^I; S)$.*

# 4 LARGE-WIDTH FUNCTIONAL LIMITS

## 4.1 LIMIT ON $C(\mathbb{R}^I; S)$, WITH $(S, d) = (\mathbb{R}, |\cdot|)$, FOR A FIXED UNIT $i \geq 1$ AND LAYER $l$

**Lemma 1** (finite-dimensional limit). *If $\phi$ satisfies (4) then there exists a stochastic process $f_i^{(l)} : \mathbb{R}^I \to \mathbb{R}$ such that $(f_i^{(l)}(n))_{n \geq 1} \xrightarrow{f_d} f_i^{(l)}$ as $n \to \infty$.*

*Proof.* Fix $l \geq 2$ and $i \geq 1$. Fixed $k$ inputs $\mathbf{X} = [x^{(1)}, \dots, x^{(k)}]$, we show that as $n \to +\infty$

$$f_i^{(l)}(\mathbf{X}, n) \xrightarrow{d} N_k(\mathbf{0}, \Sigma(l)), \tag{7}$$

where $\Sigma(l)$ denotes the $k \times k$ covariance matrix, which can be computed through the recursion: $\Sigma(1)_{i,j} = \sigma_b^2 + \sigma_\omega^2 \langle x^{(i)}, x^{(j)} \rangle_{\mathbb{R}^I}$, $\Sigma(l)_{i,j} = \sigma_b^2 + \sigma_\omega^2 \int \phi(f_i)\phi(f_j)q^{(l-1)}(\mathrm{d}f)$, where $q^{(l-1)} = N_k(\mathbf{0}, \Sigma(l-1))$. By means of (2), (3), (5) and (6),

$$\begin{cases} f_i^{(1)}(\mathbf{X}) \stackrel{d}{=} N_k(\mathbf{0}, \Sigma(1)), \quad \Sigma(1)_{i,j} = \sigma_b^2 + \sigma_\omega^2 \langle x^{(i)}, x^{(j)} \rangle_{\mathbb{R}^I} \\ f_i^{(l)}(\mathbf{X}, n)|f_{1,\dots,n}^{(l-1)} \stackrel{d}{=} N_k(\mathbf{0}, \Sigma(l, n)), \quad \text{for } l \geq 2, \\ \Sigma(l, n)_{i,j} = \sigma_b^2 + \frac{\sigma_\omega^2}{n}\left\langle (\phi \bullet \mathbf{F}_i^{(l-1)}(\mathbf{X}, n)), (\phi \bullet \mathbf{F}_j^{(l-1)}(\mathbf{X}, n)) \right\rangle_{\mathbb{R}^n} \end{cases} \tag{8}$$

We prove (7) using Levy's theorem, that is the point-wise convergence of the sequence of characteristic functions of (8). We defer to SM A for the complete proof. $\qquad \square$

Lemma 1 proves **Step A**. This proof gives an alternative and self-contained proof of the main result of Matthews et al. (2018b), under the more general assumption that the activation function $\phi$ satisfies the polynomial envelop (4). Now we prove **Step B**, i.e. the existence of the stochastic processes $f_i^{(l)}(n)$ and $f_i^{(l)}$ on the space $\mathbb{R}^{\mathbb{R}^I}$, for each layer $l \geq 1$, unit $i \geq 1$ and $n \in \mathbb{N}$. In SM E.1 we show that the finite-dimensional distributions of $f_i^{(l)}(n)$ satisfies Daniell-Kolmogorov criterion (Kallenberg, 2002, Theorem 6.16), and hence the stochastic process $f_i^{(l)}(n)$ exists. In SM E.2 we prove a similar result for the finite-dimensional distributions of the limiting process $f_i^{(l)}$. In SM E.3 we prove that, if these stochastic processes are continuous, they are naturally defined in $C(\mathbb{R}^I; \mathbb{R})$. In order to prove the continuity, i.e. **Step C** note that $f_i^{(1)}(x) = \sum_{j=1}^I \omega_{i,j}^{(1)}x_j + b_i^{(1)}$ is continuous by construction, thus by induction on $l$, if $f_i^{(l-1)}(n)$ are continuous for each $i \geq 1$ and $n$, then $f_i^{(l)}(x, n) = \frac{1}{\sqrt{n}}\sum_{j=1}^n \omega_{i,j}^{(l)}\phi(f_j^{(l-1)}(x, n)) + b_i^{(l)}$ is continuous being composition of continuous functions. For the limiting process $f_i^{(l)}$ we assume $\phi$ to be Lipschitz with Lipschitz constant $L_\phi$. In particular we have the following:

**Lemma 2** (continuity). *If $\phi$ is Lipschitz on $\mathbb{R}$ then $f_i^{(l)}(1), f_i^{(l)}(2), \dots$ are $\mathbb{P}$-a.s. Lipschitz on $\mathbb{R}^I$, while the limiting process $f_i^{(l)}$ is $\mathbb{P}$-a.s. continuous on $\mathbb{R}^I$ and locally $\gamma$-Hölder continuous for each $0 < \gamma < 1$.*

*Proof.* Here we present a sketch of the proof, and we defer to SM B.1 and SM B.2 for the complete proof. For $(f_i^{(l)}(n))_{n\geq 1}$ it is trivial to show that for each $n$

$$|f_i^{(l)}(x,n) - f_i^{(l)}(y,n)| \leq H_i^{(l)}(n)\|x-y\|_{\mathbb{R}^I}, \quad x,y \in \mathbb{R}^I, \mathbb{P} - a.s. \tag{9}$$

where $H_i^{(l)}(n)$ denotes a suitable random variable, which is defined by the following recursion over $l$

$$\begin{cases} H_i^{(1)}(n) = \sum_{j=1}^{I} |\omega_{i,j}^{(1)}| \\ H_i^{(l)}(n) = \frac{L_\phi}{\sqrt{n}} \sum_{j=1}^{n} |\omega_{i,j}^{(l)}| H_j^{(l-1)}(n) \end{cases} \tag{10}$$

To establish the continuity of the limiting process $f_i^{(l)}$ we rely on Proposition 2. Take two inputs $x,y \in \mathbb{R}^I$. From (7) we get that $[f_i^{(l)}(x), f_i^{(l)}(y)] \sim N_2(\mathbf{0}, \Sigma(l))$ where

$$\Sigma(1) = \sigma_b^2 \begin{bmatrix} 1 & 1 \\ 1 & 1 \end{bmatrix} + \sigma_\omega^2 \begin{bmatrix} \|x\|_{\mathbb{R}^I}^2 & \langle x,y \rangle_{\mathbb{R}^I} \\ \langle x,y \rangle_{\mathbb{R}^I} & \|y\|_{\mathbb{R}^I}^2 \end{bmatrix},$$

$$\Sigma(l) = \sigma_b^2 \begin{bmatrix} 1 & 1 \\ 1 & 1 \end{bmatrix} + \sigma_\omega^2 \int \begin{bmatrix} |\phi(u)|^2 & \phi(u)\phi(v) \\ \phi(u)\phi(v) & |\phi(v)|^2 \end{bmatrix} q^{(l-1)}(\mathrm{d}u, \mathrm{d}v),$$

where $q^{(l-1)} = N_2(\mathbf{0}, \Sigma(l-1))$. Defining $\mathbf{a}^T = [1,-1]$, from (7) we know that $f_i^{(l)}(y) - f_i^{(l)}(x) \sim N(\mathbf{a}^T\mathbf{0}, \mathbf{a}^T\Sigma(l)\mathbf{a})$. Thus

$$|f_i^{(l)}(y) - f_i^{(l)}(x)|^{2\theta} \sim |\sqrt{\mathbf{a}^T\Sigma(l)\mathbf{a}} N(0,1)|^{2\theta} \sim (\mathbf{a}^T\Sigma(l)\mathbf{a})^\theta |N(0,1)|^{2\theta}.$$

We proceed by induction over the layers. For $l=1$,

$$\mathbb{E}\left[|f_i^{(1)}(y) - f_i^{(1)}(x)|^{2\theta}\right] = C_\theta (\mathbf{a}^T\Sigma(1)\mathbf{a})^\theta$$
$$= C_\theta (\sigma_\omega^2 \|y\|_{\mathbb{R}^I}^2 - 2\sigma_\omega^2 \langle y,x \rangle_{\mathbb{R}^I} + \sigma_\omega^2 \|x\|_{\mathbb{R}^I}^2)^\theta$$
$$= C_\theta (\sigma_\omega^2)^\theta (\|y\|_{\mathbb{R}^I}^2 - 2\langle y,x \rangle_{\mathbb{R}^I} + \|x\|_{\mathbb{R}^I}^2)^\theta$$
$$= C_\theta (\sigma_\omega^2)^\theta \|y-x\|_{\mathbb{R}^I}^{2\theta},$$

where $C_\theta = \mathbb{E}[|N(0,1)|^{2\theta}]$. By hypothesis induction there exists a constant $H^{(l-1)} > 0$ such that $\int |u-v|^{2\theta} q^{(l-1)}(\mathrm{d}u, \mathrm{d}v) \leq H^{(l-1)} \|y-x\|_{\mathbb{R}^I}^{2\theta}$. Then,

$$|f_i^{(l)}(y) - f_i^{(l)}(x)|^{2\theta} \sim |N(0,1)|^{2\theta} (\mathbf{a}^T\Sigma(l)\mathbf{a})^\theta$$
$$= |N(0,1)|^{2\theta} \left(\sigma_\omega^2 \int [|\phi(u)|^2 - 2\phi(u)\phi(v) + |\phi(v)|^2] q^{(l-1)}(\mathrm{d}u, \mathrm{d}v)\right)^\theta$$
$$\leq |N(0,1)|^{2\theta} (\sigma_\omega^2 L_\phi^2)^\theta \int |u-v|^{2\theta} q^{(l-1)}(\mathrm{d}u, \mathrm{d}v)$$
$$\leq |N(0,1)|^{2\theta} (\sigma_\omega^2 L_\phi^2)^\theta H^{(l-1)} \|y-x\|_{\mathbb{R}^I}^{2\theta}.$$

where we used $|\phi(u)|^2 - 2\phi(u)\phi(v) + |\phi(v)|^2 = |\phi(u) - \phi(v)|^2 \leq L_\phi^2 |u-v|^2$ and the Jensen inequality. Thus,

$$\mathbb{E}\left[|f_i^{(l)}(y) - f_i^{(l)}(x)|^{2\theta}\right] \leq H^{(l)} \|y-x\|_{\mathbb{R}^I}^{2\theta}, \tag{11}$$

where the constant $H^{(l)}$ can be explicitly derived by solving the following system

$$\begin{cases} H^{(1)} = C_\theta (\sigma_\omega^2)^\theta \\ H^{(l)} = C_\theta (\sigma_\omega^2 L_\phi^2)^\theta H^{(l-1)}. \end{cases} \tag{12}$$

It is easy to get $H^{(l)} = C_\theta^l (\sigma_\omega^2)^{l\theta} (L_\phi^2)^{(l-1)\theta}$. Observe that $H^{(l)}$ does not depend on $i$ (this will be helpful in establishing the uniformly tightness of $(f_i^{(l)}(n))_{n\geq 1}$ and the continuity of $\mathbf{F}^{(l)}$). By Proposition 2, setting $\alpha = 2\theta$, and $\beta = 2\theta - I$ (since $\beta$ needs to be positive, it is sufficient to choose $\theta > I/2$) we get that $f_i^{(l)}$ has a continuous version and the latter is $\mathbb{P}$-a.s locally $\gamma$-Hölder continuous for every $0 < \gamma < 1 - \frac{I}{2\theta}$, for each $\theta > I/2$. Taking the limit as $\theta \to +\infty$ we conclude the proof. $\qquad\square$

**Lemma 3** (uniform tightness). *If $\phi$ is Lipschitz on $\mathbb{R}$ then $(f_i^{(l)}(n))_{n \geq 1}$ is uniformly tight in $C(\mathbb{R}^I; \mathbb{R})$.*

*Proof.* We defer to SM B.3 for details. Fix $i \geq 1, l \geq 1$. We apply Proposition 3 to show the uniform tightness of the sequence $(f_i^{(l)}(n))_{n \geq 1}$ in $C(\mathbb{R}^I; \mathbb{R})$. By Lemma 2 $f_i^{(l)}(1), f_i^{(l)}(2), \ldots$ are random elements in $C(\mathbb{R}^I; \mathbb{R})$. Since $(\mathbb{R}, | \cdot |)$ is Polish, every probability measure is tight, then $f(0_{\mathbb{R}^I}, n)$ is tight in $\mathbb{R}$ for every $n$. Moreover, by Lemma 1 $f_i(0_{\mathbb{R}^I}, n)_{n \geq 1} \xrightarrow{d} f_i^{(l)}(0_{\mathbb{R}^I})$, therefore by (Dudley, 2002, Theorem 11.5.3), $f(0_{\mathbb{R}^I}, n)_{n \geq 1}$ is uniformly tight in $\mathbb{R}$.

It remains to show that there exist two values $\alpha > 0$ and $\beta > 0$, and a constant $H^{(l)} > 0$ such that

$$\mathbb{E}\left[ |f_i^{(l)}(y, n) - f_i^{(l)}(x, n)|^\alpha \right] \leq H^{(l)} \|y - x\|_{\mathbb{R}^I}^{I+\beta}, \quad x, y \in \mathbb{R}^I, n \in \mathbb{N}$$

uniformly in $n$. Take two points $x, y \in \mathbb{R}^I$. From (8) we know that $f_i^{(l)}(y, n)|f_{1,\ldots,n}^{(l-1)} \sim N(0, \sigma_y^2(l, n))$ and $f_i^{(l)}(x, n)|f_{1,\ldots,n}^{(l-1)} \sim N(0, \sigma_x^2(l, n))$ with joint distribution $N_2(\mathbf{0}, \Sigma(l, n))$, where

$$\Sigma(1) = \begin{bmatrix} \sigma_x^2(1) & \Sigma(1)_{x,y} \\ \Sigma(1)_{x,y} & \sigma_y^2(1) \end{bmatrix}, \quad \Sigma(l) = \begin{bmatrix} \sigma_x^2(l, n) & \Sigma(l, n)_{x,y} \\ \Sigma(l, n)_{x,y} & \sigma_y^2(l, n) \end{bmatrix},$$

with,

$$\begin{cases} \sigma_x^2(1) = \sigma_b^2 + \sigma_\omega^2 \|x\|_{\mathbb{R}^I}^2, \\ \sigma_y^2(1) = \sigma_b^2 + \sigma_\omega^2 \|y\|_{\mathbb{R}^I}^2, \\ \Sigma(1)_{x,y} = \sigma_b^2 + \sigma_\omega^2 \langle x, y \rangle_{\mathbb{R}^I}, \\ \sigma_x^2(l, n) = \sigma_b^2 + \frac{\sigma_\omega^2}{n} \sum_{j=1}^n |\phi \circ f_j^{(l-1)}(x, n)|^2, \\ \sigma_y^2(l, n) = \sigma_b^2 + \frac{\sigma_\omega^2}{n} \sum_{j=1}^n |\phi \circ f_j^{(l-1)}(y, n)|^2, \\ \Sigma(l, n)_{x,y} = \sigma_b^2 + \frac{\sigma_\omega^2}{n} \sum_{j=1}^n \phi(f_j^{(l-1)}(x, n))\phi(f_j^{(l-1)}(y, n)) \end{cases}$$

Defining $\mathbf{a}^T = [1, -1]$ we have that $f_i^{(l)}(y, n)|f_{1,\ldots,n}^{(l-1)} - f_i^{(l)}(x, n)|f_{1,\ldots,n}^{(l-1)}$ is distributed as $N(\mathbf{a}^T \mathbf{0}, \mathbf{a}^T \Sigma(l, n)\mathbf{a})$, where $\mathbf{a}^T \Sigma(l, n)\mathbf{a} = \sigma_y^2(l, n) - 2\Sigma(l, n)_{x,y} + \sigma_x^2(l, n)$. Consider $\alpha = 2\theta$ with $\theta$ integer. Thus

$$\left| f_i^{(l)}(y, n)|f_{1,\ldots,n}^{(l-1)} - f_i^{(l)}(x, n)|f_{1,\ldots,n}^{(l-1)} \right|^{2\theta} \sim |\sqrt{\mathbf{a}^T \Sigma(l, n)\mathbf{a}} N(0, 1)|^{2\theta} \sim (\mathbf{a}^T \Sigma(l, n)\mathbf{a})^\theta |N(0, 1)|^{2\theta}.$$

As in previous theorem, for $l = 1$ we get $\mathbb{E}\left[ |f_i^{(1)}(y, n) - f_i^{(1)}(x, n)|^{2\theta} \right] = C_\theta(\sigma_\omega^2)^\theta \|y - x\|_{\mathbb{R}^I}^{2\theta}$ where $C_\theta = \mathbb{E}[|N(0, 1)^{2\theta}|]$. Set $H^{(1)} = C_\theta(\sigma_\omega^2)^\theta$ and by hypothesis induction suppose that for every $j \geq 1$

$$\mathbb{E}\left[ |f_j^{(l-1)}(y, n) - f_j^{(l-1)}(x, n)|^{2\theta} \right] \leq H^{(l-1)} \|y - x\|_{\mathbb{R}^I}^{2\theta}.$$

By hypothesis $\phi$ is Lipschitz, then

$$\begin{aligned}
\mathbb{E}\left[ |f_i^{(l)}(y, n) - f_i^{(l)}(x, n)|^{2\theta} \Big| f_{1,\ldots,n}^{(l-1)} \right] &= C_\theta(\mathbf{a}^T \Sigma(l, n)\mathbf{a})^\theta \\
&= C_\theta\left( \sigma_y^2(l, n) - 2\Sigma(l, n)_{x,y} + \sigma_x^2(l, n) \right)^\theta \\
&= C_\theta\left( \frac{\sigma_\omega^2}{n} \sum_{j=1}^n \left| \phi \circ f_j^{(l-1)}(y, n) - \phi \circ f_j^{(l-1)}(x, n) \right|^2 \right)^\theta \\
&\leq C_\theta\left( \frac{\sigma_\omega^2 L_\phi^2}{n} \sum_{j=1}^n \left| f_j^{(l-1)}(y, n) - f_j^{(l-1)}(x, n) \right|^2 \right)^\theta \\
&= C_\theta \frac{(\sigma_\omega^2 L_\phi^2)^\theta}{n^\theta} \left( \sum_{j=1}^n \left| f_j^{(l-1)}(y, n) - f_j^{(l-1)}(x, n) \right|^2 \right)^\theta \\
&\leq C_\theta \frac{(\sigma_\omega^2 L_\phi^2)^\theta}{n} \sum_{j=1}^n \left| f_j^{(l-1)}(y, n) - f_j^{(l-1)}(x, n) \right|^{2\theta}.
\end{aligned}$$

Using the induction hypothesis

$$
\begin{aligned}
\mathbb{E}\Big[|f_i^{(l)}(y,n) - f_i^{(l)}(x,n)|^{2\theta}\Big] &= \mathbb{E}\Big[\mathbb{E}\Big[|f_i^{(l)}(y,n) - f_i^{(l)}(x,n)|^{2\theta}\Big|f_{1,\ldots,n}^{(l-1)}\Big]\Big] \\
&\leq C_\theta \frac{(\sigma_\omega^2 L_\phi^2)^\theta}{n} \sum_{j=1}^n \mathbb{E}\Big[|f_j^{(l-1)}(y,n) - f_j^{(l-1)}(x,n)|^{2\theta}\Big] \\
&\leq C_\theta (\sigma_\omega^2 L_\phi^2)^\theta H^{(l-1)} \|y - x\|_{\mathbb{R}^I}^{2\theta}.
\end{aligned}
$$

We can get the constant $H^{(l)}$ by solving the same system as (12), obtaining $H^{(l)} = C_\theta^l (\sigma_\omega^2)^{l\theta}(L_\phi^2)^{(l-1)\theta}$ which does not depend on $n$. By Proposition 3 setting $\alpha = 2\theta$ and $\beta = 2\theta - I$, since $\beta$ must be a positive constant, it is sufficient to take $\theta > I/2$ and this concludes the proof. $\square$

Note that Lemma 3 provides the last **Step D** that allows us to prove the desired result which is explained in the theorem that follows:

**Theorem 1** (functional limit). *If $\phi$ is Lipschitz on $\mathbb{R}$ then $f_i^{(l)}(n) \overset{d}{\to} f_i^{(l)}$ on $C(\mathbb{R}^I; \mathbb{R})$.*

*Proof.* We apply Proposition 1 to $(f_i^{(l)}(n))_{n\geq1}$. By Lemma 2, we have that $f_i^{(l)}, (f_i^{(l)}(n))_{n\geq1}$ belong to $C(\mathbb{R}^I; \mathbb{R})$. From Lemma 1 we have the convergence of the finite-dimensional distributions of $(f_i^{(l)}(n))_{n\geq1}$, and form Lemma 3 we have the uniform tightness of $(f_i^{(l)}(n))_{n\geq1}$. $\square$

## 4.2 LIMIT ON $C(\mathbb{R}^I; S)$, WITH $(S, d) = (\mathbb{R}^\infty, \|\cdot\|_{\mathbb{R}^\infty})$, FOR A FIXED LAYER $l$

As in the previous section we prove Steps A-D for the sequence $(\mathbf{F}^{(l)}(n))_{n\geq1}$. Remark that each stochastic process $\mathbf{F}^{(l)}, \mathbf{F}^{(l)}(1), \mathbf{F}^{(l)}(2), \ldots$ defines on $C(\mathbb{R}^I; \mathbb{R}^\infty)$ a joint measure whose $i$-th marginal is the measure induced respectively by $f_i^{(l)}, f_i^{(l)}(n), f_i^{(2)}(n), \ldots$ (see SM E.1 -SM E.4). Let $\mathbf{F}^{(l)} \overset{d}{=} \bigotimes_{i=1}^\infty f_i^{(l)}$, where $\bigotimes$ denotes the product measure.

**Lemma 4** (finite-dimensional limit). *If $\phi$ satisfies (4) then $\mathbf{F}^{(l)}(n) \overset{f_d}{\to} \mathbf{F}^{(l)}$ as $n \to \infty$.*

*Proof.* The proof follows by Lemma 1 and Cramér-Wold theorem for finite-dimensional projection of $\mathbf{F}^{(l)}(n)$: it is sufficient to establish the large $n$ asymptotic of linear combinations of the $f_i^{(l)}(\mathbf{X}, n)$'s for $i \in \mathcal{L} \subset \mathbb{N}$. In particular, we show that for any choice of inputs elements $\mathbf{X}$, as $n \to +\infty$

$$
\mathbf{F}^{(l)}(\mathbf{X}, n) \overset{d}{\to} \bigotimes_{i=1}^\infty N_k(\mathbf{0}, \Sigma(l)), \tag{13}
$$

where $\Sigma(l)$ is defined in (7). The proof is reported in SM C. $\square$

**Lemma 5** (continuity). *If $\phi$ is Lipschitz on $\mathbb{R}$ then $\mathbf{F}^{(l)}, (\mathbf{F}^{(l)}(n))_{n\geq1}$ belong to $C(\mathbb{R}^I; \mathbb{R}^\infty)$. More precisely $\mathbf{F}^{(l)}(1), \mathbf{F}^{(l)}(2), \ldots$ are $\mathbb{P}$-a.s. Lipschitz on $\mathbb{R}^I$, while the limiting process $\mathbf{F}^{(l)}$ is $\mathbb{P}$-a.s. continuous on $\mathbb{R}^I$ and locally $\gamma$-Hölder continuous for each $0 < \gamma < 1$.*

*Proof.* It derives immediately from Lemma 2. We defer to SM D.1 and SM D.2 for details. The continuity of the sequence process immediately follows from the Lipschitzianity of each component in (9) while the continuity of the limiting process $\mathbf{F}^{(l)}$ is proved by applying Proposition 2. Take two inputs $x, y \in \mathbb{R}^I$ and fix $\alpha \geq 1$ even integer. Since $\xi(t) \leq t$ for all $t \geq 0$, and by Jensen inequality

$$
d\big(\mathbf{F}^{(l)}(x), \mathbf{F}^{(l)}(y)\big)_\infty^\alpha \leq \Big(\sum_{i=1}^\infty \frac{1}{2^i}|f_i^{(l)}(x) - f_i^{(l)}(y)|\Big)^\alpha \leq \sum_{i=1}^\infty \frac{1}{2^i}|f_i^{(l)}(x) - f_i^{(l)}(y)|^\alpha
$$

Thus, by applying monotone convergence theorem to the positive increasing sequence $g(N) = \sum_{i=1}^N \frac{1}{2^i}|f_i^{(l)}(x) - f_i^{(l)}(y)|^\alpha$ (which allows to exchange $\mathbb{E}$ and $\sum_{i=1}^\infty$), we get

$$\mathbb{E}\Big[d\big(\mathbf{F}^{(l)}(x),\mathbf{F}^{(l)}(y)\big)_\infty^\alpha\Big] \le \mathbb{E}\Big[\sum_{i=1}^\infty \frac{1}{2^i}|f^{(l)}(x)-f_i^{(l)}(y)|^\alpha\Big] = \lim_{N\to\infty}\mathbb{E}\Big[\sum_{i=1}^N \frac{1}{2^i}|f_i^{(l)}(x)-f_i^{(l)}(y)|^\alpha\Big]$$

$$= \sum_{i=1}^\infty \frac{1}{2^i}\mathbb{E}\Big[|f_i^{(l)}(x)-f_i^{(l)}(y)|^\alpha\Big] = \sum_{i=1}^\infty \frac{1}{2^i}H^{(l)}\|x-y\|_{\mathbb{R}^I}^\alpha = H^{(l)}\|x-y\|_{\mathbb{R}^I}^\alpha$$

where we used (11) and the fact that $H^{(l)}$ does not depend on $i$ (see (12)). Therefore, by Proposition 2, for each $\alpha > I$, setting $\beta = \alpha - I$ (since $\beta$ needs to be positive, it is sufficient to choose $\alpha > I$) $\mathbf{F}^{(l)}$ has a continuous version $\mathbf{F}^{(l)(\theta)}$ which is $\mathbb{P}$-a.s locally $\gamma$-Hölder continuous for every $0 < \gamma < 1 - \frac{I}{\alpha}$. Letting $\alpha \to \infty$ we conclude. $\qquad\square$

**Theorem 2** (functional limit). *If $\phi$ is Lipschitz on $\mathbb{R}$ then $(\mathbf{F}^{(l)}(n))_{n\ge 1} \xrightarrow{d} \mathbf{F}^{(l)}$ as $n \to \infty$ on $C(\mathbb{R}^I;\mathbb{R}^\infty)$.*

*Proof.* This is Proposition 1 applied to $(\mathbf{F}^{(l)}(n))_{n\ge 1}$. From Lemma 4 and Lemma 5 it remains to show the uniform tightness of the sequence $(\mathbf{F}^{(l)}(n))_{n\ge 1}$ in $C(\mathbb{R}^I;\mathbb{R}^\infty)$. Let $\epsilon > 0$ and let $(\epsilon_i)_{i\ge 1}$ be a positive sequence such that $\sum_{i=1}^\infty \epsilon_i = \epsilon/2$ . We have established the uniform tightness of each component (Lemma 3). Therefore for each $i \in \mathbb{N}$ there exists a compact $K_i \subset C(\mathbb{R}^I;\mathbb{R})$ such that $\mathbb{P}[f_i^{(l)}(n) \in C(\mathbb{R}^I;\mathbb{R}) \setminus K_i] < \epsilon_i$ for each $n \in \mathbb{N}$ (such compact depends on $\epsilon_i$). Set $K = \times_{i=1}^\infty K_i$ which is compact by Tychonoff theorem. Note that this is a compact on the product space $\times_{i=1}^\infty C(\mathbb{R}^I;\mathbb{R})$ with associated product topology, and this is also a compact on $C(\mathbb{R}^I;\mathbb{R}^\infty)$ (see SM E.4). Then $\mathbb{P}\Big[\mathbf{F}^{(l)}(n) \in C(\mathbb{R}^I;\mathbb{R}^\infty) \setminus K\Big] = \mathbb{P}\Big[\bigcup_{i=1}^\infty \{f_i^{(l)}(n) \in C(\mathbb{R}^I;\mathbb{R}) \setminus K_i\}\Big] \le \sum_{i=1}^\infty \mathbb{P}\Big[f_i^{(l)}(n) \in C(\mathbb{R}^I;\mathbb{R}) \setminus K_i\Big] \le \sum_{i=1}^\infty \epsilon_i < \epsilon$ which concludes the proof. $\qquad\square$

## 5 DISCUSSION

We looked at deep Gaussian neural networks as stochastic processes, i.e. infinite-dimensional random elements, on the input space $\mathbb{R}^I$, and we showed that: i) a network defines a stochastic process on the input space $\mathbb{R}^I$; ii) under suitable assumptions on the activation function, a network with re-scaled weights converges weakly to a Gaussian Process in the large-width limit. These results extend previous works (Neal, 1995; Der & Lee, 2006; Lee et al., 2018; Matthews et al., 2018a;b; Yang, 2019) that investigate the limiting distribution of neural network over a countable number of distinct inputs. From the point of view of applications, the convergence in distribution is the starting point for the convergence of expectations. Let consider a continuous function $g : C(\mathbb{R}^I;\mathbb{R}^\infty) \to \mathbb{R}$. By the continuous function mapping theorem (Billingsley, 1999, Theorem 2.7), we have $g(\mathbf{F}^{(l)}(n)) \xrightarrow{d} g(\mathbf{F}^{(l)})$ as $n \to +\infty$, and under uniform integrability (Billingsley, 1999, Section 3), we have (Billingsley, 1999, Theorem 3.5) $\mathbb{E}[g(\mathbf{F}^{(l)}(n))] \to \mathbb{E}[g(\mathbf{F}^{(l)})]$ as $n \to +\infty$. See also Dudley (2002) and references therein.

As a by-product of our results we showed that, under a Lipschitz activation function, the limiting Gaussian Process has almost surely locally $\gamma$-Hölder continuous paths, for $0 < \gamma < 1$. This raises the question on whether it is possible to strengthen our results to cover the case $\gamma = 1$, or even the case of local Lipschitzianity of the paths of the limiting process. In addition, if the activation function is differentiable, does this property transfer to the limiting process? We leave these questions to future research. Finally, while fully-connected deep neural networks represent an ideal starting point for theoretical analysis, modern neural network architectures are composed of a much richer class of layers which includes convolutional, residual, recurrent and attention components. The technical arguments followed in this paper are amenable to extensions to more complex network architectures. Providing a mathematical formulation of network's architectures and convergence results in a way that it allows for extensions to arbitrary architectures, instead of providing an ad-hoc proof for each specific case, is a fundamental research problem. Greg Yang's work on Tensor Programs (Yang, 2019) constitutes an important step in this direction.

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
