# OpenReview forum: "Large-width functional asymptotics for deep Gaussian neural networks"
_ICLR.cc/2021/Conference — ICLR 2021 Poster_

### Official Review · AnonReviewer1 · 2020-10-28
**I suggest the acceptance but I still question the significance**

**Rating:** 6
**Confidence:** 4

**Review:**

**Summary**:
The paper is based on recent research about the limit of neural networks to Gaussian processes. The main result is the proof of the limit from a function perspective: neural networks are considered as random processes with the input space being infinite-dimensional.

In the beginning, I didn’t see the motivation for doing another proof for the limit. However, in the discussion section, the authors pointed to the possible convergence of expectations where their approach can be used. Also I think the setting is different from the setting of the previous result: their NNs are of the width the same for each layer instead of strictly increasing (Matthews et al, 2018). I didn’t notice flaws in the proofs so if I'm not mistaken the authors provided proof for a bit more general statement.  But I'm still not sure about its significance.

Overall, I find the paper well written and not difficult to follow. There are some inaccuracies which are easy to fix.

**Comments**:
- “Neal (1995) showed that for each $l \ge 2$ and a fixed unit $i$, the $k$-dimensional random vector $(f^{(l)}_i(x^{1}), \dots, f^{(l)}_i(x^{k}))$ converges in distribution, as the widths $n_1, \dots , n_L$ go to infinity sequentially over network layers.“
I think that Neal showed that under certain conditions random neural networks with *one* hidden layer converge to a Gaussian process. This statement was proved by Lee et al. (2018).

- “We operate in the same setting, hence from here onward n 1 denotes the common layer width, i.e. $n_1, \dots, n_L = n$.”
Actually, it is not the same setting as Matthews et al. (2018). Their theorem states the existence of a *strictly increasing* width function such that the NN distribution convergence to a GP.

**Minor**:
- Infinitely wide -> infinitely-wide
- full/fully connected -> fully-connected
- Finite dimensional - finite-dimensional
- Network’s layers -> network layers
- “Of a neural networks”
- “Such an approach is competitive with the more standard stochastic gradient descent training for the fully- connected architectures object of our study”
- “Shows that the for Gaussian Processes…”
- “We start by proving the existence of: i)” I think the colon is unnecessary
- “the limiting Gaussian process have”
- Section 2, “k-dimensional Gaussian distribution with mean \mu…” \mu should be bold because before (\mu was 1-dimensional). Moreover, I think it would be better to use m instead because \mu is used as a measure notation later
- “the characteristic function of b_i , for for”
- Definition 1, eq. (6): first equality is between identical expressions; “a vector of dimension k \times 1 of 1s” - should be of “1’s”; “x_j denote” -> denotes; IS the element-wise…
- Definition 2: the expressional after “where” was not used in the equations before. I think there should be firstly everything after the word “vector” and then the rest.
- “Since applying the Definition 3” - should be without “the”
- “It remain to show”
- “Finally, while fully connected deep neural networks represents”
- References: the same authors have different names and “Gaussian” should be capital

---

> ### Author Response · Authors · 2020-11-17
> **Reply to AnonReviewer1**
>
> We thank the Reviewer and address his/her main comments:
>
> > “Neal (1995) showed that for each  and a fixed unit , the -dimensional random vector  converges in distribution, as the widths  go to infinity sequentially over network layers.“ I think that Neal showed that under certain conditions random neural networks with one hidden layer converge to a Gaussian process. This statement was proved by Lee et al. (2018).
>
> Please see our reply to AnonReviewer2 regarding this same comment. We have modified the text to reflect a more correct attribution.
>
> > “We operate in the same setting, hence from here onward n 1 denotes the common layer width, i.e.” Actually, it is not the same setting as Matthews et al. (2018). Their theorem states the existence of a strictly increasing width function such that the NN distribution convergence to a GP.
>
> Our statement was unclear: convergence over an increasing sub-sequence was established in the ICLR paper, and the results had been extended to cover the general case in the expanded arXiv version. We removed this ambiguity in the updated text.
>
> We also addressed all the minor issues raised (not reported here), and thank once again the Reviewr for identifying all the typos. We remain available for further clarifications in case any point were not addressed in a satisfactory manner.

---

### Official Review · AnonReviewer2 · 2020-10-29
**Good paper but need clarification about some proof steps; paper structure could be improved**

**Rating:** 7
**Confidence:** 3

**Review:**

POST-REBUTTAL

Thank you for your answers and incorporated modifications! I think you've succeeded in addressing my major concerns, so I'm raising my score and recommending accept as promised.

---

This paper is a theoretical investigation of the asymptotic behaviour of deep fully connected neural networks (FCNs) as the number of hidden units in each layer goes to infinity (simultaneously in all layers). Its goal is to extend the existing results of Neal (1995), Matthews et al. (2018), Lee & Bahri et al. (2018), and others (see Yang, 2019, for an overview), from countable to uncountable index sets (input space). This is typically done by proving convergence of finite marginals, and establishing relative compactness of the sequence of processes induced here by the finite networks. As the former was already done by Matthews et al., Yang, and others, the main contribution of this paper is the study of relative compactness. The authors approach this problem by recognising that with additional Lipschitz assumption on the nonlinearity, both the finite network induced processes and the asymptotic GP limit have a version with paths in a space of continuous mapping between two Polish spaces (at least in most of typical use cases). This is then helpful because the space of functions between a compact separable space and a Polish space is itself Polish (under the sup-norm topology), and the links between weakly convergent and uniformly tight sequences on Polish spaces.

Overall, I think the topic of this paper is a good fit for ICLR (most of the new work in this direction was presented at ICLR in the last three years), and that the paper is trying to address an important open question. The reason I have assigned a relatively low score for now is because I’m unsure about the validity of some steps in the proofs, and because I think the structure of the paper could improve. However, I am hopeful that the authors may be able to satisfactorily address (at least) my “major comments” during the rebuttal period, in which case I would improve my score to 7 or 8.


MAJOR COMMENTS

* In the proof of Theorem 2, you construct K as the Cartesian product of compact sets for each of the infinitely many hidden unit marginals (nit: may want to mention this is compact by Tychnoff’s theorem). I am not sure whether I’m misunderstanding the notation or something else, but the penultimate line of the proof looks a bit suspect. IIUC, F^\ell (n) will be in K^c if **at least one** (not necessarily all) of the countably many units f_i^\ell (n) is outside of its K_i, so one should be taking union over the {f_i^\ell(n) \in K_i^c} events, not an intersection?! The rest of the proof doesn’t work when this replacement is made though. Could you please clarify?

* It might be useful to add some discussion (e.g., into the appendix) of the involved sigma-algebras. In particular, you cite theorem 6.16 from Kallenberg (Kolmogorov extension theorem) which only guarantees existence of the given stochastic process on the product sigma-algebra. On the other hand, the theorems from chapter 16 of Kallenberg AFAICT apply to stochastic processes on the Borel sigma-algebra generated on C(R^I; S) by the sup-norm; I think that if you replaced R^I by a separable compact set, these two sigma-algebras will coincide, but I’m not sure this is true with R^I itself?! Isn’t the Borel sigma-algebra generated by the sup norm strictly larger with the whole R^I? If so, could you please clarify why this is not an issue?

* On page 4, you say C(R^I; S) is a subspace of C(K; S) where K is a (compact) subset of R^I, which you need to invoke proposition 16.4 of Kallenberg. I suppose I could see how C(R^I; S) is isomorphic to a subspace of C(K; S), but that doesn’t seem to be what you mean here? If it is, could you please provide more details on how this would be sufficient for the conclusion of Proposition 1? If not, could you clarify please?

* Since “convergence on uncountable index set = convergence on finite sets + relative compactness”, I am not sure if the decision to spend ~2 out of the 8 pages on rederiving a known result by Matthews et al. is the best use of space. It’s absolutely ok (even commendable) to desire to be self-contained, but the space in the main text may be better spent explaining the novel material on relative compactness/tightness, and the Holder continuity (the proofs & discussion of which are largely relegated to the appendix), and moving the alternative derivation of Matthews et al.’s results into the appendix instead.

* Relatedly, in the introduction you say “we include a compact proof of the main result of Matthews et al.” implying their proof is significantly longer/more complicated. However, the two proofs share many steps, as the main difference is that the weak convergence of Cramer-Wald projections is proved via Levy’s continuity theorem in your case, and via the exchangeable CLT (Blum et al., 1958) by Matthews et al. Each of these two methods requires checking three conditions (your L1-L3; a-c in Lemma 10 in Matthews et al.) which itself involves many of the similar steps (uniform integrability via uniform control over moments, inductive propagation of weak convergence through layers, etc.), and without the additional commentary injected by Matthews et al., approximately the same amount of space. I would suggest replacing the word “compact” with “alternative” in the above statement. Furthermore, I think it would benefit the reader if you acknowledged both in the main tex  and in the appendix the parts of your proof based on/inspired by Matthews et al.:
  - Exploiting exchangeability is one of the main ideas of Matthews et al., albeit they apply the exchangeable CLT whereas you use the Levy’s continuity theorem (which is nice!). Matthews et al. also propagate weak convergence and uniform integrability across the layers in the proof of their proposition 14 (cf. your Lemma 2 in the appendix).
  - Your construction in Equations (5) and (6) is the ‘infinite width, finite fan-out, network’ construction from Matthews et al.
  - The use of Cramer-Wold theorem is also in Matthews et al. (which you mention in the appendix, but not on p. 8).
  - The polynomial envelope condition is a straightforward generalisation of Matthews et al. “linear envelope”, and the proof of Matthews would work with it as well (the only change would be in propagating uniform boundedness of higher than 8th moments in their lemma 20). Also, your proof of Lemma 1 in the appendix is quite similar to Matthews et al.’s lemma 20 (combined with lemma 19). As an aside, your Lipschitzness assumption implies |phi(x)| <= |phi(0)| + L |x|, i.e., your later developments confine the discussion to a subset of the “linear envelope” satisfying activations from Matthews et al.
  - Relatedly, Novak & Xiao & Lee et al. (2018) and also Yang (2019; 2020) have already shown requiring polynomially bounded nonlinearity is sufficient which should be mentioned as well.

* It has been a while since I read Radford Neal’s PhD thesis, but I cannot recall a proof of the \ell > 2 “sequential limit” you ascribe to him on page 1 (in particular, I do not think he studied theoretically single-hidden layer FCNs with inputs drawn from a GP). AFAIK this work was done over several papers by Hazan & Jaakola, Lee & Bahri et al., and others. Could you please clarify?


MINOR COMMENTS

* I am wondering about the extension to NTK which can---at least in the limit---be viewed as a linear transformation of the prior distribution (if f is a sample from the NNGP prior, and one assumes MSE loss, a sample from NTK evaluated at x is equal in distribution to f(x) + Theta(x, X) (y - f(X)), where (X, y) is the training set, and Theta is the neural tangent kernel). Do you think your results translate also to gradient descent trained FCNs? Relatedly, there is a recent paper called “Exact posterior distributions of wide Bayesian neural networks” which you may want to cite; it shows that when the prior converges, posterior converges too. The proof is quite short and should apply to your setup (i.e., uncountable index set) without changes.

* Am I right that the assumption of the same width of each hidden layer is not necessary?

* I think the proof of L1 requires some discussion of uniform integrability. Overall, it might be best to have a separate lemma establishing uniform integrability of relevant random variables as currently the reader looks at L1-L3, and thinks that your proof is incorrect (convergence in probability generally doesn’t imply convergence of expectation without uniform integrability). Uniform integrability (mentioned only in passing within the proof of L1) is also utilised in SM A.1.4 which should be highlighted.

* Can you comment on measurability of the event Y_s from Lemma 1? Are you working with the Borel sigma-algebra of the weak* topology, or is this meant as more of a shorthand?!

* In the proof of L3 (p.14, appendix), is the supremum over n supposed to be there?

* On p.4, step B), I think you only establish that both stochastic processes have versions in C(R^I; S) here, but rather only in step C).

* On p.6, the first display after L3 (the 2-line expression for the characteristic function), it would help readability if the parenthesis around the expression raised to the nth power were braces (or at least brackets), and larger.

* I am somewhat confused by the sentence “convergence in distribution is the starting point for the convergence of expectations” in your conclusion. I didn’t check your proofs carefully enough to be 100% sure, but AFAICT you can relatively easily combine the established uniform integrability and weak convergence to get convergence of (at least finite-marginal) mean and covariance (as you seem to be aware). Do you mean expectations of other non-bounded or non-continuous functions of the output?

* In several places, including the abstract and the conclusion, you seem to say that each **finite** network defines a **Gaussian** process which converges in distribution to another Gaussian process as n->infinity. Did you mean to say that the networks define a **stochastic** instead of specifically a Gaussian process? (I don’t think that finite networks generally induce a Gaussian process, nor was this proved.)

* In several places you claim that Matthews et al. and Neal only proved convergence on finite spaces, and only sometimes mention that the results were proved also for countable index sets. I think the reader would benefit if you always said countable, as not everyone knows that the extension to countable spaces is relatively straightforward consequence (in the same way as weak convergence on all or R^I is a relatively straightforward consequence of convergence on all compact subsets---proposition 16.6 in Kallenberg).

* In the last sentence of your conclusion, you mention as a future direction development of a “mathematical formulation of network’s architectures and convergence results in a way that it allows for extensions to arbitrary architectures”. You are right that this framework is missing for the exchangeability based proof techniques, but the reader would benefit from contextualising this statement with respect to Greg Yang’s work on Tensor Programs.

---

> ### Author Response · Authors · 2020-11-17
> **Reply to AnonReviewer2 - Part I**
>
> We wish to thank the Reviewer for his/her very thorough and detailed review of the present paper. Such a review clearly took a significant amount of time and greatly contributed to improving the quality of this work. We now address all the major concerns raised:
>
> > In the proof of Theorem 2, you construct $K$ as the Cartesian product of compact sets for each of the infinitely many hidden unit marginals (nit: may want to mention this is compact by Tychnoff’s theorem). I am not sure whether I’m misunderstanding the notation or something else, but the penultimate line of the proof looks a bit suspect. IIUC, $F^\ell(n)$ will be in $K^c$ if at least one (not necessarily all) of the countably many units $f_i^\ell(n)$ is outside of its $K_i$, so one should be taking union over the ${f_i^\ell(n) \in K_i^c}$ events, not an intersection?! The rest of the proof doesn’t work when this replacement is made though. Could you please clarify?
>
> Indeed the proof of Theorem 2 as reported is not correct. The result still holds, and the updated text contains a revised proof of Theorem 2.
>
> > It might be useful to add some discussion (e.g., into the appendix) of the involved sigma-algebras. In particular, you cite theorem 6.16 from Kallenberg (Kolmogorov extension theorem) which only guarantees existence of the given stochastic process on the product sigma-algebra. On the other hand, the theorems from chapter 16 of Kallenberg AFAICT apply to stochastic processes on the Borel sigma-algebra generated on $C(\mathbb{R}^I; S)$ by the sup-norm; I think that if you replaced R^I by a separable compact set, these two sigma-algebras will coincide, but I’m not sure this is true with $\mathbb{R}^I$ itself?! Isn’t the Borel sigma-algebra generated by the sup norm strictly larger with the whole $\mathbb{R}^I$? If so, could you please clarify why this is not an issue?
>
> We devoted a new section in the appendix, SM E3, for a detailed discussion on this technical but important point. Parts of the main text have been modified accordingly.
>
> > On page 4, you say $C(R^I; S)$ is a subspace of $C(K; S)$ where $K$ is a (compact) subset of $\mathbb{R}^I$, which you need to invoke proposition 16.4 of Kallenberg. I suppose I could see how $C(\mathbb{R}^I; S)$ is isomorphic to a subspace of $C(K; S)$, but that doesn’t seem to be what you mean here? If it is, could you please provide more details on how this would be sufficient for the conclusion of Proposition 1? If not, could you clarify please?
>
> The proof as written was indeed quite unclear. In the revised text the proof has been moved to SM F (following AnonReviewer3 suggestion) and the use of proposition 16.4 clarified.
>
> > Since “convergence on uncountable index set = convergence on finite sets + relative compactness”, I am not sure if the decision to spend ~2 out of the 8 pages on rederiving a known result by Matthews et al. is the best use of space. It’s absolutely ok (even commendable) to desire to be self-contained, but the space in the main text may be better spent explaining the novel material on relative compactness/tightness, and the Hölder continuity (the proofs & discussion of which are largely relegated to the appendix), and moving the alternative derivation of Matthews et al.’s results into the appendix instead.
>
> > Relatedly, in the introduction you say “we include a compact proof of the main result of Matthews et al.” implying their proof is significantly longer/more complicated. However, the two proofs share many steps, as the main difference is that the weak convergence of Cramer-Wald projections is proved via Levy’s continuity theorem in your case, and via the exchangeable CLT (Blum et al., 1958) by Matthews et al. Each of these two methods requires checking three conditions (your L1-L3; a-c in Lemma 10 in Matthews et al.) which itself involves many of the similar steps (uniform integrability via uniform control over moments, inductive propagation of weak convergence through layers, etc.), and without the additional commentary injected by Matthews et al., approximately the same amount of space. I would suggest replacing the word “compact” with “alternative” in the above statement. Furthermore, I think it would benefit the reader if you acknowledged both in the main tex  and in the appendix the parts of your proof based on/inspired by Matthews et al...
>
> We essentially eliminated the proof from the main text. We kept only equations (7) and (8) detailing the form of the covariance recursion, as this is used later in the main text (Lemma 2).
>
> \[continue...\]

---

> ### Author Response · Authors · 2020-11-17
> **Reply to AnonReviewer2 - Part II**
>
> > It has been a while since I read Radford Neal’s PhD thesis, but I cannot recall a proof of the $\ell > 2$ “sequential limit” you ascribe to him on page 1 (in particular, I do not think he studied theoretically single-hidden layer FCNs with inputs drawn from a GP). AFAIK this work was done over several papers by Hazan & Jaakola, Lee & Bahri et al., and others. Could you please clarify?
>
> We were honestly unsure about how to handle merit attribution here. Neal (1995) had a section titled "Priors for networks with more than one hidden layer" where preliminary considerations were made for deep networks. These includes the use of the correct scaling ($H^{-1/2}$) for each layer and the mention of Gaussian limits. But it is true that such a result has been formally established only in later works, such as Lee et al. (2018). We have modified the introduction accordingly.
>
> We now address the minor points raised:
>
> > I am wondering about the extension to NTK which can---at least in the limit---be viewed as a linear transformation of the prior distribution (if f is a sample from the NNGP prior, and one assumes MSE loss, a sample from NTK evaluated at x is equal in distribution to f(x) + Theta(x, X) (y - f(X)), where (X, y) is the training set, and Theta is the neural tangent kernel). Do you think your results translate also to gradient descent trained FCNs?
>
> Yes, we considered in what sense, and with what arguments, the present work could be extended to the fully-trained (i.e. NTK) regime. Such an extension is however non-trivial, and we do not plan to address this (interesting!) question in the present work.
>
> > Relatedly, there is a recent paper called “Exact posterior distributions of wide Bayesian neural networks” which you may want to cite; it shows that when the prior converges, posterior converges too. The proof is quite short and should apply to your setup (i.e., uncountable index set) without changes.
>
> It would indeed be a relevant work to cite. It is not clear from the main text on what space the probability measures of Proposition 1 are defined (i.e. whether Proposition 1 concerns a single or multiple inputs, one or multiple network units). We will check the proof and cite the work once we fully understand the context.
>
> > Am I right that the assumption of the same width of each hidden layer is not necessary?
>
> It seems possible, see our reply to the same point of AnonReviewer3. However we did not check carefully yet whether there are technical impediments, we will try to come back to this point in time for the rebuttal.
>
> > I think the proof of L1 requires some discussion of uniform integrability. Overall, it might be best to have a separate lemma establishing uniform integrability of relevant random variables as currently the reader looks at L1-L3, and thinks that your proof is incorrect (convergence in probability generally doesn’t imply convergence of expectation without uniform integrability). Uniform integrability (mentioned only in passing within the proof of L1) is also utilised in SM A.1.4 which should be highlighted.
>
> We modified SM A.1.4. to clarify the use of uniform integrability. We also mentioned how uniform integrability is achieved in the places where it is used, i.e. via $L^s$-boundness.
>
> > Can you comment on measurability of the event $Y_s$ from Lemma 1? Are you working with the Borel sigma-algebra of the weak* topology, or is this meant as more of a shorthand?!
>
> We address this measurability issue at the end of SM A.1.1.
>
> > In the proof of L3 (p.14, appendix), is the supremum over n supposed to be there?
>
> It is a mistake, which we rectified in the updated text, since L1) states the $L^s$ bounded for each s.
>
> > On p.4, step B), I think you only establish that both stochastic processes have versions in $C(\mathbb{R}^I; S)$ here, but rather only in step C).
>
> We corrected the statements both in the Step A-B and in SM E.
>
> > On p.6, the first display after L3 (the 2-line expression for the characteristic function), it would help readability if the parenthesis around the expression raised to the nth power were braces (or at least brackets), and larger.
>
> We have made it more readable as suggested.
>
> > I am somewhat confused by the sentence “convergence in distribution is the starting point for the convergence of expectations” in your conclusion. I didn’t check your proofs carefully enough to be 100% sure, but AFAICT you can relatively easily combine the established uniform integrability and weak convergence to get convergence of (at least finite-marginal) mean and covariance (as you seem to be aware). Do you mean expectations of other non-bounded or non-continuous functions of the output?
>
> In the discussion we meant a continuous but not necessarily bounded function(al) $g$, in a general setting.
>
> \[continue...\]

---

> ### Author Response · Authors · 2020-11-17
> **Reply to AnonReviewer2 - Part III**
>
> > In several places, including the abstract and the conclusion, you seem to say that each finite network defines a Gaussian process which converges in distribution to another Gaussian process as n->infinity. Did you mean to say that the networks define a stochastic instead of specifically a Gaussian process? (I don’t think that finite networks generally induce a Gaussian process, nor was this proved.)
>
> Yes, the text as stated is clearly incorrect (AnonReviewer3 raised the same point). We apologize for this issue, which has been rectified in the updated text.
>
> > In several places you claim that Matthews et al. and Neal only proved convergence on finite spaces, and only sometimes mention that the results were proved also for countable index sets. I think the reader would benefit if you always said countable, as not everyone knows that the extension to countable spaces is relatively straightforward consequence (in the same way as weak convergence on all or R^I is a relatively straightforward consequence of convergence on all compact subsets---proposition 16.6 in Kallenberg).
>
> We only kept mentions of a finite number of inputs in the introduction relatively to Neal seminal work. We kept such initial discussion limited to a finite number of inputs because: i) we thought it would be a bit confusing for a very first example to be on infinite spaces ii) to the best of our knowledge results of countable nature in input space are not discussed anywhere in Neal (1995). In the remaining parts of the text we removed mentions of finite number of inputs.
>
> > In the last sentence of your conclusion, you mention as a future direction development of a “mathematical formulation of network’s architectures and convergence results in a way that it allows for extensions to arbitrary architectures”. You are right that this framework is missing for the exchangeability based proof techniques, but the reader would benefit from contextualising this statement with respect to Greg Yang’s work on Tensor Programs.
>
> We agree and we added the reference in the discussion section.
>
> We remain available for further clarifications in case any point were not addressed in a satisfactory manner.

---

### Official Review · AnonReviewer4 · 2020-10-31
**This paper presents an analysis on the large-width functional asymptotics of the fully connected feed-forward deep neural networks where weights and biases are independent and identically distributed according to Gaussian distributions and proves that such a network defines a continuous Gaussian process  with certain limiting properties.  However, I doubt  these results can be used to improve the learning performance  of the network for either general or specific task.**

**Rating:** 4
**Confidence:** 4

**Review:**

This paper makes a deep analysis on the large-width functional asymptotics of the fully connected feed-forward deep neural networks where weights and biases are independent and identically distributed according to Gaussian distributions.  Actually, it proves that this kind of network tends to be a continuous Gaussian process as the width of the layer increases to infinity. Some more specific convergence properties are also obtained along this direction. It is clear that these results are theoretically significant. But I cannot find out how they can be applied in the field of learning theory and algorithms.  If a network can model a Gaussian process with a specific setting of the wights and biases, it is significant. However,  these weights and biases are assumed to be random as noises,  the network will not be learned with any task and cannot contribute to machine learning. Therefore, I cannot recommend this paper for the conference.

---

> ### Author Response · Authors · 2020-11-17
> **Reply to AnonReviewer4**
>
> We thank the Reviewer and address the following common misunderstanding:
>
> > This paper makes a deep analysis on the large-width functional asymptotics of the fully connected feed-forward deep neural networks where weights and biases are independent and identically distributed according to Gaussian distributions. Actually, it proves that this kind of network tends to be a continuous Gaussian process as the width of the layer increases to infinity. Some more specific convergence properties are also obtained along this direction. It is clear that these results are theoretically significant. But I cannot find out how they can be applied in the field of learning theory and algorithms. If a network can model a Gaussian process with a specific setting of the weights and biases, it is significant. However, these weights and biases are assumed to be random as noises, the network will not be learned with any task and cannot contribute to machine learning. Therefore, I cannot recommend this paper for the conference.
>
> It is true that the weights and biases a priori are independent and identically distributed (iid). However this only defines the prior distribution of such weights and biases, and in turn the prior distribution in function space of the model outputs given its inputs. In the context here considered, it is then possible to perform Bayesian inference. Quoting from our main text: "Such an approach \[Bayesian inference for infinitely wide fully-connected neural networks\] is competitive with the more standard stochastic gradient descent training for the fully-connected architectures object of our study (Lee et al. 2020)". We refer to (Lee et al. 2020) for a detailed empirical study which includes the setting considered in our work. We remain available for further clarifications.

---

### Official Review · AnonReviewer3 · 2020-11-01
**A rigorous proof of DNN infinite width limits from a functional/stochastic process viewpoint**

**Rating:** 7
**Confidence:** 4

**Review:**

The paper nicely completes the picture of distributional results for infinite width limits of DNN. More specifically, the common framework in this literature proves limits for a finite or countable number of inputs (Neal 1995, Matthews et al 2018), while this paper addresses the question from a stochastic process viewpoint over the input space (in a sense, this corresponds to proving a large width result that holds jointly over an infinite number of inputs).

The paper is well-written and clear, despite its rather technical nature. I think it is original. The significance seems more debatable to me: actually, all the known results over a finite or countable number of inputs carry over to the infinite-dimensional joint number of inputs. This is somehow expected, and I'm unsure about how these new results are going to benefit the community. But to me, this slight downside does not necessarily detract from the relevance of the paper and overall aesthetics of the proofs.

Specific comments:
- $I$ is never defined as the input space dimension
- p2: "shows that the for Gaussian Processes "
- there are several instances of the sentence "proving the existence of: i) a continuous Gaussian process, indexed by the network's width n, corresponding to the fully connected feed-forward deep neural network" (with variations, in Abstract, Introduction and Discussion). This is slightly ambiguous to me, when I read it, I understand that you prove that to any NN with fixed-width n, there corresponds a GP. However, I do not think this is what you prove (ie  a fixed-width NN is not a GP).
- could you please discuss the d(a,b)_\infty distance and associated norm in more detail? I do not know it, and it does not look like trivial that is satisfies the triangle inequality. This embeds the infinite-dimensional space R^\infty with a norm. If all distances are equivalent in finite dim spaces, this is not the case here where R^\infty is infinite dim. What sort of properties should a distance/norm possess so that your results still hold true?
- the envelope condition (4) does not prevent the non-linearity to be identically constant (eg zero); just to be sure, are all of your results compatible with this possibility (I think they are thanks to the presence of the bias terms)?
- unless there is a strong motivation for it, I would not include the proof of Proposition 1 in the main text, in order to focus more closely to ML related topics of higher interest to ICLR readership.
- p5: "for any for any"
- middle of p6: \theta should be \theta_n or vice-versa.
- top of p7: should specify that \Sigma(l) matrix corresponds to the two inputs (x,y) so that it is a 2x2 matrix. Of course, this can be understood with previous notations, but this is ambiguous as is.
- p7, two line above eq (12): $L_\phi$ should be $L_\phi^2$, I think.
- p7, just before Lemma 3: "it is sufficient to choose $\theta>I$; why not choosing $\theta>I/2$, which seems ok as well?
- the assumption that all widths n_1,...,n_L are equal is unrealistic and only made for technical reasons. Could you discuss how this could be relaxed?
- the discussion section provides some insights on the implication of Lemma 5 and the \gamma-Hölderianity of the limiting process GP paths, for 0 <  \gamma < 1. Could you discuss potential implications of this type of smoothness, and also what it would provide to obtain a stronger result covering the \gamma=1 case?
- the bibliography could be tidied (capitals to gaussian)

---

> ### Author Response · Authors · 2020-11-17
> **Reply to AnonReviewer3**
>
> We wish to thank the Reviewer for his/her appraisal. We address below all the raised comments:
>
> > $I$ is never defined as the input space dimension
>
> It is defined after equation (1).
>
> > top of p7: should specify that $\Sigma(l)$ matrix corresponds to the two inputs (x,y) so that it is a 2x2 matrix. Of course, this can be understood with previous notations, but this is ambiguous as is.
>
> We agree, the proof has been updated.
>
> > p7, just before Lemma 3: "it is sufficient to choose $\theta > I$; why not choosing $\theta > I/2$, which seems ok as well?
>
> Yes, this seems clearer, we have updated the proof accordingly.
>
> > there are several instances of the sentence "proving the existence of: i) a continuous Gaussian process, indexed by the network's width n, corresponding to the fully connected feed-forward deep neural network" (with variations, in Abstract, Introduction and Discussion). This is slightly ambiguous to me, when I read it, I understand that you prove that to any NN with fixed-width n, there corresponds a GP. However, I do not think this is what you prove (ie  a fixed-width NN is not a GP).
>
> We apologize for statement i) which is definitely not correct, this has been rectified in the updated text.
>
> > could you please discuss the $d(a,b)_\infty$ distance and associated norm in more detail? I do not know it, and it does not look like trivial that is satisfies the triangle inequality. This embeds the infinite-dimensional space $R^\infty$ with a norm. If all distances are equivalent in finite dim spaces, this is not the case here where $R^\infty$ is infinite dim. What sort of properties should a distance/norm possess so that your results still hold true?
>
> We added a reference (Charalambos D. Aliprantis, Kim C. Border - Infinite-dimensional analysis-Springer (2006)) to show that $d(a,b)_\infty$ as defined is a distance on $\mathbb{R}^{\infty}$. Moreover, as mentioned in the proof of Theorem 3.36 of this same reference, $d(a,b)_\infty$ induces the product topology.
>
> > the envelope condition (4) does not prevent the non-linearity to be identically constant (eg zero); just to be sure, are all of your results compatible with this possibility (I think they are thanks to the presence of the bias terms)?
>
> Yes, all results continue to hold as we are assuming strictly positive variances for the parameters ($\mathbb{R}_+$ does not include 0 in our notation, we made this point explicit in the updated text) and due to the last layer definition.
>
> > unless there is a strong motivation for it, I would not include the proof of Proposition 1 in the main text, in order to focus more closely to ML related topics of higher interest to ICLR readership.
>
> We moved this proof in SM F.
>
> > the assumption that all widths n_1,...,n_L are equal is unrealistic and only made for technical reasons. Could you discuss how this could be relaxed?
>
> We agree that it would be more realistic to establish our results under the weaker condition $\min{n_1,...,n_L} \rightarrow \infty$. We did not have the time to work through all the proofs in detail to verify whether major adjustments are required or not. We will try to extend our results prior to the rebuttal deadline time permitting.
>
> > the discussion section provides some insights on the implication of Lemma 5 and the \gamma-Hölderianity of the limiting process GP paths, for 0 < \gamma < 1. Could you discuss potential implications of this type of smoothness, and also what it would provide to obtain a stronger result covering the $\gamma=1$ case?
>
> In the main text we provided one possible application, quoting: "For instance, VanDer Vaart & Van Zanten (2011)  shows that for Gaussian Processes the function smoothness under the prior should match the smoothness of the target function for satisfactory inference performance." Establishing $\gamma=1$ is a requisite for establishing the differentiability of the limiting paths, i.e. stronger smoothness properties.
>
> > the bibliography could be tidied (capitals to gaussian)
>
> There seems to be an issue with the ICLR 2021 bibliography style file, as article titles get converted to lower case. We are investigating how to fix this issue.
>
> We also fixed all the mentioned typos (thanks!) which we did not explicitly include in the above list. We remain available for further clarifications in case any point were not addressed in a satisfactory manner.

---

### Author Response · Authors · 2020-11-23
**Updated paper**

To avoid any misunderstanding: the paper (main text and supplementary material) has been updated in line with our replies on the 17th.

---

### Decision · Program_Chairs · 2021-01-07
**Final Decision**

**Decision:**

Accept (Poster)

**Comment:**

This article provides an analysis of feedforward neural network with iid Gaussian weights and biases in the infinite-width limit. The paper  complements earlier work on this topic by taking a function-space approach, considering neural networks as infinite-dimensional random elements on the input space. This is a well-written and rigorous theoretical paper. Although, as noted by a reviewer, there are no direct practical implications, the result is interesting in itself, highly relevant to the ICLR audience, and likely to lead to further exploration of the connections between Gaussian processes and neural networks.

There were a few questions regarding the proofs that have been answered satisfactorily by the authors.

I recommend acceptance.